# Cryo-EM structures and binding of mouse and human ACE2 to SARS-CoV-2 variants of concern indicate that mutations enabling immune escape could expand host range

**Dongchun Ni**[1,2], **Priscilla Turelli**[3], **Bertrand Beckert**[4], **Sergey Nazarov**[4], **Emiko Uchikawa**[4], **Alexander Myasnikov**[4], **Florence Pojer**[5], **Didier Trono**[3], **Henning Stahlberg**[1,2]*, **Kelvin Lau**[5]*

**1** Laboratory of Biological Electron Microscopy (LBEM), Institute of Physics, School of Basic Science, École Polytechnique Fédérale de Lausanne (EPFL), Lausanne, Switzerland, **2** Dep. of Fund. Microbiology, Faculty of Biology and Medicine, University of Lausanne, Lausanne, Switzerland, **3** Laboratory of Virology and Genetics (LVG), School of Life Sciences, École polytechnique Fédérale de Lausanne (EPFL), Lausanne, Switzerland, **4** Dubochet Center for Imaging (DCI), École polytechnique Fédérale de Lausanne (EPFL) and University of Lausanne, Lausanne, Switzerland, **5** Protein Production and Structure Characterization Core Facility (PTPSP), School of Life Sciences, École polytechnique Fédérale de Lausanne (EPFL), Lausanne, Switzerland

* henning.stahlberg@epfl.ch (HS); kelvin.lau@epfl.ch (KL)

**Data Availability Statement:** DATA AVAILABILITY Cryo-EM maps for the Spike variants in complex with mouse ACE2 were deposited in the Electron Microscopy Data Bank (EMDB) under the access

## Abstract

Investigation of potential hosts of the severe acute respiratory syndrome coronavirus-2 (SARS-CoV-2) is crucial to understanding future risks of spillover and spillback. SARS-CoV-2 has been reported to be transmitted from humans to various animals after requiring relatively few mutations. There is significant interest in describing how the virus interacts with mice as they are well adapted to human environments, are used widely as infection models and can be infected. Structural and binding data of the mouse ACE2 receptor with the Spike protein of newly identified SARS-CoV-2 variants are needed to better understand the impact of immune system evading mutations present in variants of concern (VOC). Previous studies have developed mouse-adapted variants and identified residues critical for binding to heterologous ACE2 receptors. Here we report the cryo-EM structures of mouse ACE2 bound to trimeric Spike ectodomains of four different VOC: Beta, Omicron BA.1, Omicron BA.2.12.1 and Omicron BA.4/5. These variants represent the oldest to the newest variants known to bind the mouse ACE2 receptor. Our high-resolution structural data complemented with bio-layer interferometry (BLI) binding assays reveal a requirement for a combination of mutations in the Spike protein that enable binding to the mouse ACE2 receptor.

## Author summary

The SARS-CoV-2 virus can infect different types of animals beyond humans. The virus uses its Spike protein on its surface to bind to cells. These cells have a protein called ACE2 that the Spike protein recognizes. Animals have slightly different ACE2 receptors

codes EMD-15541 (full map, Beta, two mACE2 bound), EMD-15589 (local map, mACE2/Beta), EMD- 15580 (full map, BA.1, two mACE2 bound), EMD-15581 (full map, BA.1, three mACE2 bound), EMD-15590 (local map, mACE2/BA.1), EMD-15584 (full map, BA.2.12.1, two mACE2 bound), EMD-15585 (full map, BA2.12.1, three mACE2 bound), EMD-15591 (local map, mACE2/ BA.2.12.1), EMD-15586 (full map, BA.4/5, two mACE2 bound) and EMD-15592 (local map, mACE2/BA.4/5). Maps for Spike the human ACE2/ Omicron BA.4/5 complex were deposited in the EMDB under the access codes EMD-15587 (full map, BA.4/5, three hACE2 bound) and EMD-15588 (local map, hACE2/BA.4/5). Atomic models were deposited in Protein Data Bank (PDB) under the access codes of PDB-8AQS (https://www.rcsb.org/structure/8AQS) (hACE2-BA.4/5), PDB-8AQT (https://www.rcsb.org/structure/8AQT) (mACE2/Beta), PDB-8AQU (https://www.rcsb.org/structure/8AQU) (mACE2/BA.1), PDB-8AQV (https://www.rcsb.org/structure/8AQV) (mACE2/BA.2.12.1) and PDB-8AQW (https://www.rcsb.org/structure/8AQW)(mACE2/BA.4/5). Raw electron microscopy image data were deposited at the Electron Microscopy Public Image Archive (EMPIAR) under access codes EMPIAR-11181 (mACE2/Beta), EMPIAR-11179 (mACE2/BA.1), EMPIAR-11180 (mACE2/BA.2.12.1), EMPIAR-11177 (mACE2/ BA.4/5), EMPIAR-11176 (hACE2/BA.4/5).

**Funding:** D.N and H.S acknowledge funding by an NCCR (National Centre of Competence in Research) TransCure grant (185544) from the Schweizerischer Nationalfonds zur Förderung der Wissenschaftlichen Forschung (Swiss National Science Foundation). K.L and F.P were supported by core funding provided by the EPFL School of Life Sciences (SV). The funders had no role in the the study design, data collection and analysis, decision to publish or preparation of the manuscript.

**Competing interests:** The authors declare no competing interests in regards to this work.

compared to humans. Mice are widely used as a research animal and live in the same environment as humans so scientists are particularly interested. Understanding how Spike proteins bind to the mouse ACE2 receptor allows us to understand the impact of immune evading mutations found in new variants. We use a high resolution imaging technique called cryo-electron microscopy to look at how different Spike variants bind to the ACE2 receptor from mouse at a resolution where we can see the amino acids. We can see directly the individual amino acids and mutations on the Spike protein that interact with the mouse ACE2 receptor. Many of the mutations found in variants of concern also increase the strength of binding to the mouse ACE2 receptor. This result suggests that mutations in the Spike protein of future variants may have an additional effect in influencing how it binds to not only human ACE2 receptors but to mice and also different animals.

## Introduction

Severe Acute Respiratory Syndrome (SARS) is a zoonotic disease caused by SARS-CoV infection, and its symptoms can range from mild to severe.[1] SARS-CoV was first identified in 2003 and it sparked a major global public health crisis.[2,3] Covid-19 is caused by SARS-CoV-2 infection, which was first reported in Wuhan in late 2019 and has led to a global pandemic. [4–6] Both viruses are of animal origin and their intermediate hosts prior to the jump to humans have not been precisely identified, however, it is accepted that like the SARS-CoV virus, the original host of the SARS-CoV-2 virus are most likely Rhinophilus bats.[7–10] It is well understood that angiotensin-converting enzyme 2 (ACE2) is a major receptor for host cell invasion by a range of coronaviruses, including SARS-CoV-2.[11–15] The Spike glycoprotein embedded in the viral envelope binds specifically to human or animal ACE2 on the cell surface, where it undergoes further membrane fusion and ultimately enters and infects the host cell. Due to the conserved nature of ACE2 in animals and constant selection of Spike variants, particularly ones that evade the human immune system, SARS-CoV-2 may exhibit a high degree of spillover and spillback.[16–19] Several animal infections or exposures have occurred during the covid-19 outbreak, including cats, dogs, rats, minks and American white-tailed deer.[20–23]

SARS-CoV-2 evades the human immune system in several ways, such as accumulating mutations on the Spike protein that reduce binding to host antibodies.[24–28] Significantly, many sites of mutations in VOC are located on the receptor binding domain (RBD) that engages ACE2. Mutations that endow the ability for the Spike protein to evade antibodies in humans may also play an additional role by influencing ACE2 binding specificity and host range. The availability of 3D structures of variant RBDs interacting with the receptor is crucial for understanding how they continue to bind not only human ACE2 but potentially those of other animals. Indeed, there has been speculation and critical debate that the Omicron variant could have arisen in a mouse reservoir.[29–32] There is interest in describing the interactions between mouse ACE2 and the Spike protein, as it is a commonly used infection model and there has been work done with mouse-adapted strains.[33–35] There have been structural studies of animal ACE2 receptors with the SARS-CoV-2 Spike protein but few focusing on mouse across many different variants.[36–42] To better inform the relationship between these variants and potential species tropism, there is a need to examine the structural details between animal ACE2 receptors and the variant Spike proteins.

In this study, using cryo-electron microscopy single particle analysis (cryo-EM SPA) we resolved a series of high-resolution structures of the SARS-CoV-2 variant Spikes in complex with mouse ACE2, including the Beta, BA.1, BA.2.12.1 and BA.4/5 variants. We compared the binding affinity of different variants Spike to the receptors. We further analyzed the high-resolution structure of the widespread variant, BA.4/5, with human ACE2 using cryo-EM. Together, these suggest that there is a level of structural plasticity in the RBD of the Spike protein that is implicated in both adaptation for animal hosts and immune escape.

## Results

### Mouse ACE2 binds to Spike variants of concern Alpha, Beta, Gamma and Omicron subvariants

We performed a BLI binding screen between SARS-CoV-2 variants of concern Alpha, Beta and Gamma against a set of animal ACE2, including mouse ACE2 (mACE2), compared to human ACE2 (hACE2). Interestingly, unlike cat, dog and mink ACE2 that could bind to the wild-type Spike protein, mACE2 only showed a substantial binding signal against these VOC, similar to other reports.[19,43,44] (S1 Fig) We decided to focus on and better characterize mACE2 binding affinities against these Spike VOC and performed full BLI binding assays at three concentrations. The wild-type Spike protein bound robustly hACE2 but not strongly to mACE2, with only appreciable signal at the highest concentration of 300 nM (Fig 1). The emergence of the N501Y mutation found in the receptor binding domain (RBD) of the Spike protein of Alpha, Beta and Gamma variants (S2 Fig) has been associated with mice susceptibility to infection.[45–48] In accordance with these results we found that Alpha, Beta and Gamma Spikes had a higher affinity ($K_d$ between 35, 45 and 25 nM, respectively) for mACE2 compared to the ancestral Spike protein ($K_d$ above 100nM) (Fig 1A). From the end of 2021, Omicron strains have caused successive waves of infections worldwide.[49] While affinity for hACE2 of Omicron BA.1 and BA.2 Spikes was 2.5 times increased compared to the ancestral Spike, affinity for mACE2 was increased more than 40 times (Fig 1A). The Spike protein from the more recent Omicron BA.2.12.1 and BA.4/5 variants, containing respectively 1 and 3 mutations in the RBD compared to BA.2, retain a strong affinity for mACE2 (Figs 1A and S2).

### Cryo-EM structures of Spike variants of concern bound to mouse ACE2 and BA.4/5 to human ACE2

To probe how variants of concern gained high affinity mACE2 binding we used cryo-EM to determine the structures of trimeric Spike variants in complex with human and mouse ACE2 receptors. We used Spikes from Beta, Omicron BA.1, BA.2.12.1, and BA.4/5, and obtained resolutions of 3.9 Å, 2.7Å, 2.9 Å and 2.9Å, respectively (Figs 2 and S3–S7). We also solved the human ACE2/BA.4/5 Spike complex at 2.8Å for comparison (Fig 3). Our cryo-EM structures determined with dimeric ACE2-Fc and a trimerized Spike ectodomain recapitulates the hACE2/wild-type complexes observed in other studies done by x-ray crystallography (PDB 6M0J) or cryo-EM (PDB 7A94), with monomeric ACE2 and RBD or monomeric ACE2 and trimerized Spike, respectively. All of our structures show that mACE2/hACE2 bind the RBD of the Spike protein in the same manner at the binding interface of a RBD in the up-position. The RBD is highly mobile and its transition between the down- to its up- state reveals the ACE2 binding site and allows for it to bind ACE2. The process is dynamic, and the proportion of RBD-down and -up has been shown to be dependent on many factors such as mutations, glycans, pH and temperature.[50–57] During cryo-EM data-processing we observed all the Spikes with a mixture of 2-up or 3-up conformation, with each up-RBD having an ACE2

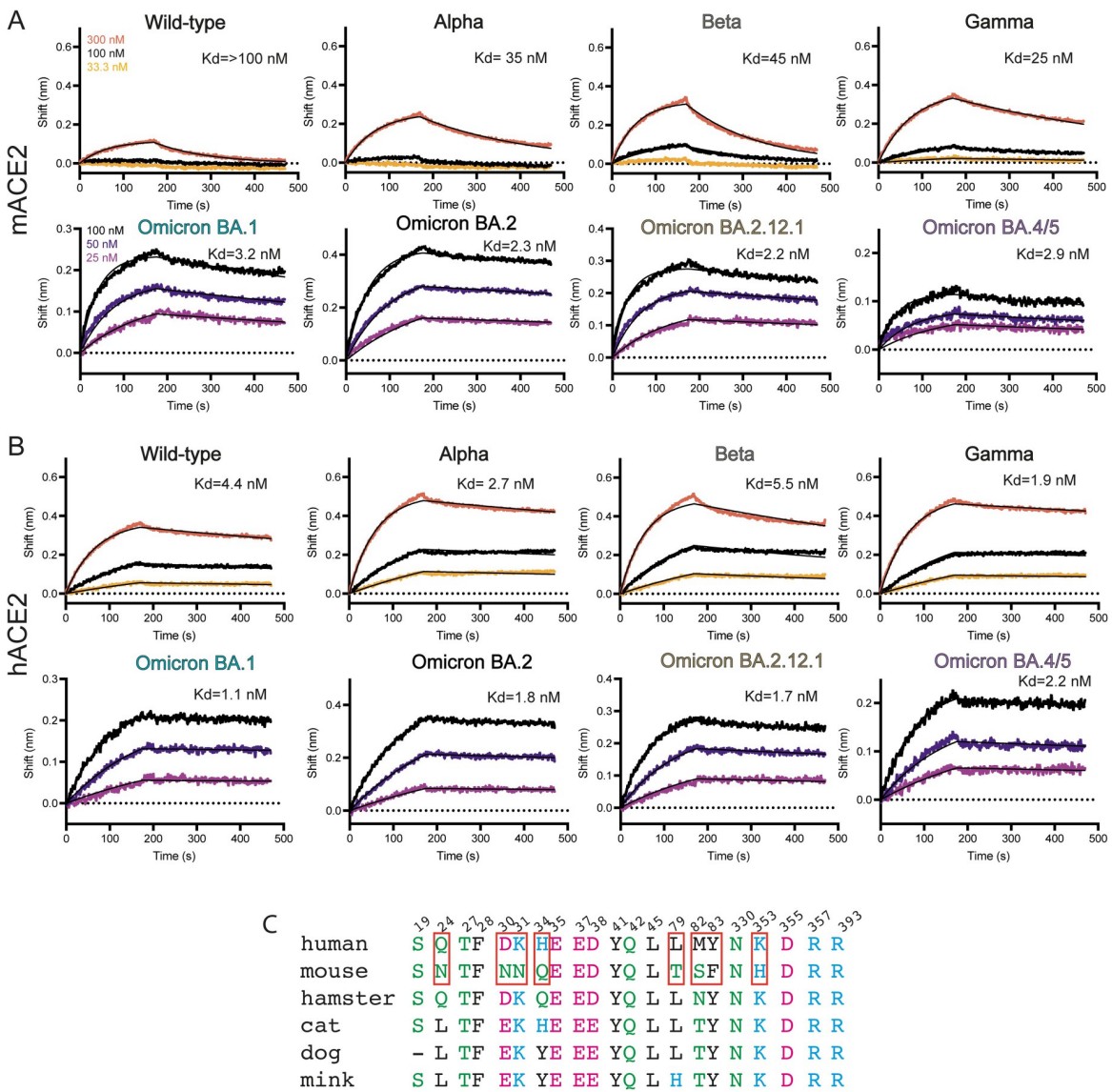

**Fig 1. mACE2 and hACE2 binding to variants of concern. (A)** BLI binding assays of captured dimeric mouse ACE2 versus various concentrations of Spike variants of concern. Data curves are colored by concentration and the black line indicates the 1:1 fit of the data. **(B)** BLI binding assays of captured dimeric human ACE2 versus various concentrations of Spike variants of concern. Data curves are colored by concentration and the black line indicates the 1:1 fit of the data. **(C)** Sequence alignment of human ACE2 with mouse ACE2 and other selected species. Red boxes highlight critical differences between human and mouse ACE2 residues.

bound (S3–S7 Figs). To gain structural information at side-chain level resolution of the binding interface, we performed image processing of the cryo-EM data by location-focused refinement on an ACE2-bound RBD that was the most well resolved. This yielded maps at higher quality for each variant's ACE2/RBD interface, allowing us to better understand the effects of local mutations in these regions (Figs 2B and 3 and S3–S8). Each of the mACE2/RBD complex structures were highly superimposable, in particular the RBD domain and the two helices of mACE2 that interact with it, with 0.5–0.8 Å RMSD compared to the mACE2/BA.1 complex (Fig 2C). The hACE2/BA.4/5 complex is also highly superimposable with that of the crystal structure of the isolated wildtype RBD in complex with monomeric hACE2 (PDB: 6M0J) with an RMSD of 0.9 Å and with movements principally at the ACE2 lobe distal to the interaction

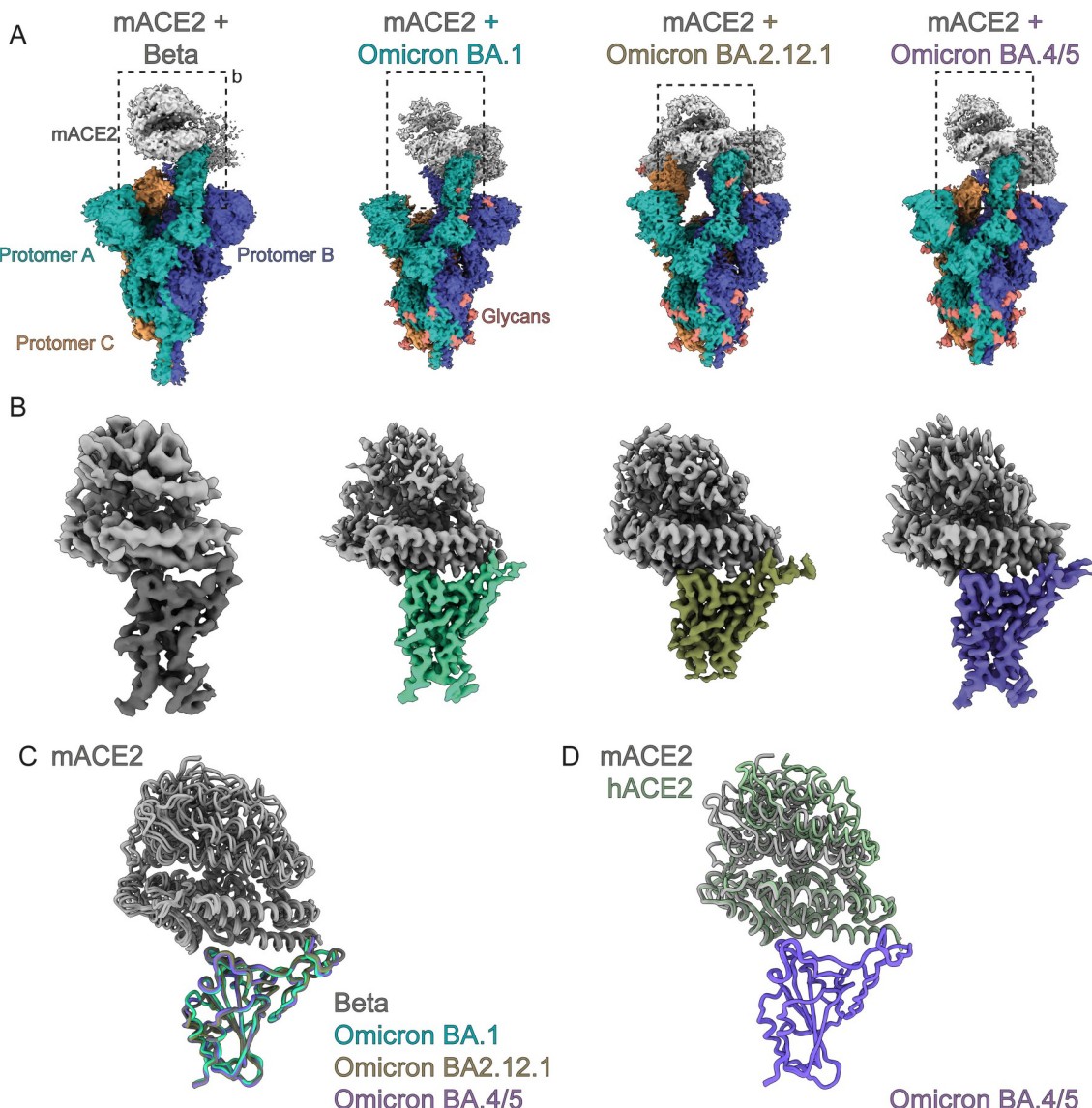

**Fig 2. Cryo-EM structures of mACE2 bound to variants of concern.** (A) Cryo-EM densities of the full mACE2/Spike variant of concern complexes. Each protomer of the Spike trimer is colored separately with mACE2 colored in grey. (B) Focused refinement of the RBD-mACE2 interface of each complex as in (A). (C) Superposition of RBD-mACE2 complexes on to the BA.1-mACE2 complex show mACE2 binds similarly for all variants. (D) Superposition of the BA.4/5 RBD-hACE2 complex with the BA.4/5 RBD-mACE2 complex.

site with the RBD (Fig 3B). The hACE2/BA.4/5 complex is very similar to the mACE2/BA.4/5 complex with an overall RMSD of 0.6 Å (Fig 2D). We were also able to build most mutations and residues present in the variants highlighting the high-quality of our cryo-EM reconstructions (Figs 4 and S9).

## The combination of the N501Y and E484K mutation allow for Spike binding to mACE2

The first mutation reported to be crucial for mACE2 engagement and shared by the Alpha, Beta and Gamma and the more recent Omicron variants is the N501Y mutation. This

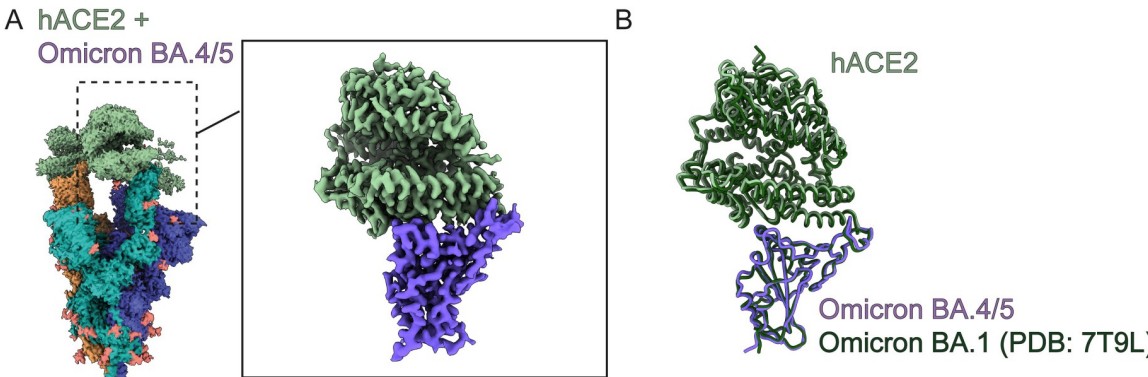

**Fig 3. Cryo-EM structures of hACE2 bound to the BA.4/5 Spike. (A)** Cryo-EM densities of the full hACE2/BA.4/5 Spike. Each protomer of the Spike trimer is colored separately with hACE2 colored in light green. Inset shows focused refinement of the RBD-hACE2 interface. **(B)** Superposition of BA.4/5 RBD-hACE2 complex with the BA.1 RBD-hACE2 complex (PDB 7T9L).

mutation is located within the RBD and is one residue that has been identified to enable higher affinity interactions with the mACE2 receptor.[35,58,59] In the mACE2/Beta structure, we observe that the Spike N501Y mutation which replaces a polar asparagine (N) residue with a polar and aromatic tyrosine (Y) residue, allows for π-π interactions with Y41 of mACE2, and enables a new cation-π and hydrogen bond interaction with histidine H353 of mACE2 (Figs 1C and 4B). These new interactions stabilize the interface between mACE2 and the Beta RBD. Beta and Gamma both also share the E484K and K417N/T substitutions in the Spike RBD (S2 Fig). These two residues are unfortunately not resolved in our cryo-EM maps. However, *in vitro* BLI experiments showed that while the introduction of either N501Y, K417N or the E484K mutations alone are not sufficient to confer binding to mACE2, at the concentration of Spike protein tested here (75 nM), the combination of Y501 and K484 led to appreciable binding of the mutant Spike to mACE2 (S10 Fig).

## BA.1 mutations Q493R and Q498R further increase binding to mACE2

We next investigated the impact of additional mutations found in Omicron variants BA.1, BA.2, BA.2.12.1 and BA.4/5. These variants contain up to 34 mutations scattered across the entire Spike protein, with the 15 of those concentrated on the RBD, including N501Y and E484A (S2 Fig). In particular, some Omicron RBD mutations (K417N, G446S(BA.1), E484A, F486S (BA.4/5) Q493R, G496S(BA.1), Q498R (BA.1, BA.2, BA.2.12.1) and N501Y) have been shown to be associated with immune evasion, with a loss of antibody binding or higher affinity ACE2 binding.[28,60–63] The accumulation of these mutations at once prompted an ongoing discussion whether the Omicron variant may have arisen in a chronically infected COVID-19 person who is immunocompromised, or possibly spilled over from a mouse reservoir.[31] Indeed, the mACE2/BA.1 structure reveals multiple new interactions localized in two different patches between the RBD and mACE2 (Fig 4A and 4B). In patch 1, the longer arginine (R) sidechain of the BA.1 Q493R mutation allows for new hydrogen bonds with mACE2 residues N31 and Q34. These were not possible with the shorter glutamine (Q) residue present in the original Spike protein. In patch 2, Y501 is observed to form cation-π interactions with H353 from mACE2, as in the mACE2/Beta complex, and new BA.1 mutations G496S and Q498R form a new set of hydrogen bonds and electrostatic interactions with mACE2's aspartic acid D38 (Fig 4B). This also reorients Y449 of the Omicron BA.1 RBD to form a new hydrogen bond with Q42 of mACE2. Furthermore, mutation of Y505 to a smaller histidine residue now allows for better packing at the interface and potentially a CH-π T-shape interaction.[64]

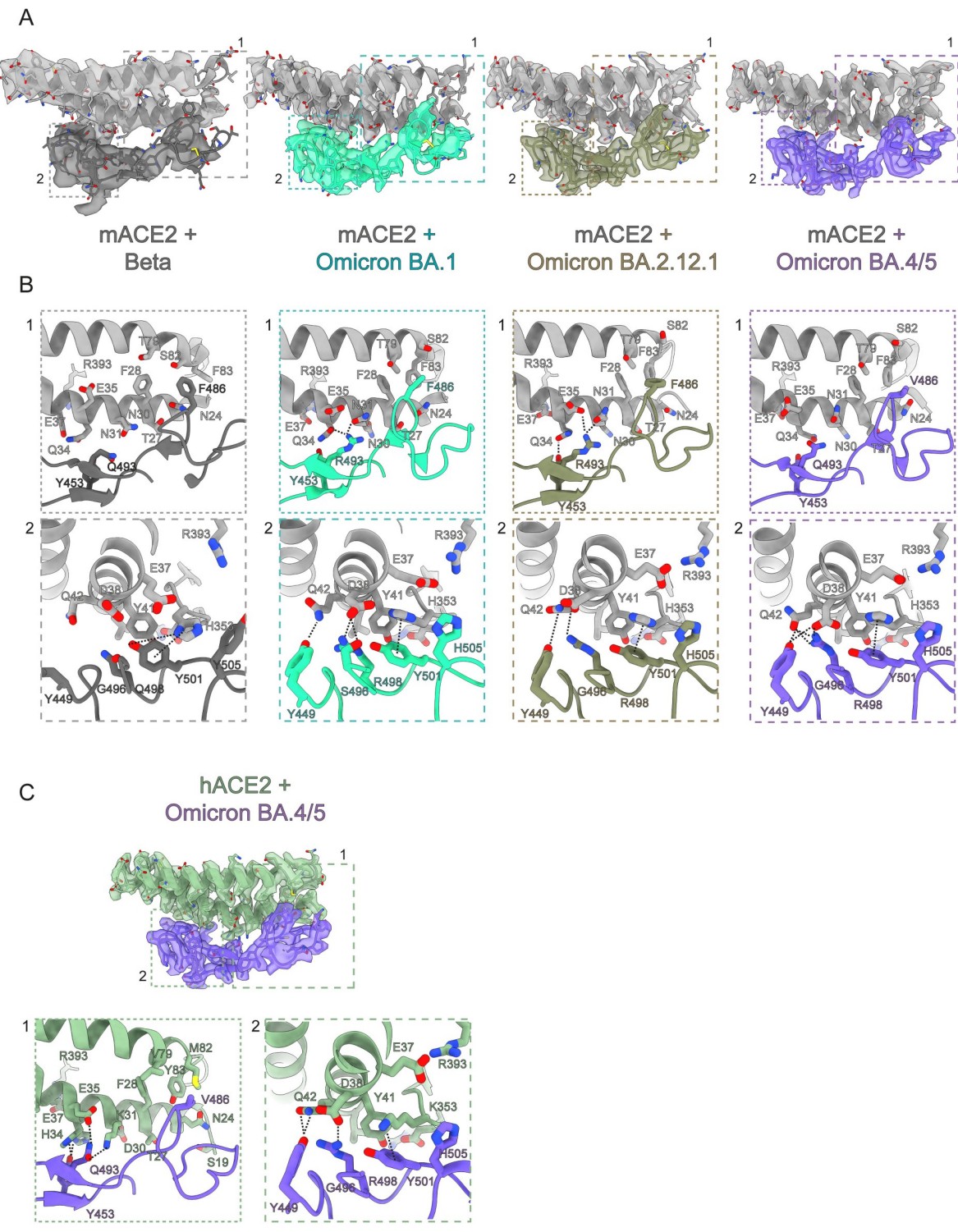

**Fig 4. Structural basis of mACE2 binding to variants of concern and of hACE2/BA.4/5. (A)** Zoomed view of the binding interface between mACE2 and RBD of variants of concern with cryo-EM densities. **(B)** Highlighted views of specific interaction sites of patch 1 and patch 2 as indicated in (A). **(C)** Zoomed view of the binding interface between hACE2 and RBD of variants of concern with cryo-EM densities with highlighted views of specific interaction sites.

Glutamate (E) 37 also flips away from the interface and is now sandwiched between H353 and R393, stabilizing the overall conformation of the interface. Overall, Omicron BA.1 mutations Q493R and Q498R form a cascade of new interactions alongside the N501Y mutation present in preceding variants. These additional contacts greatly increase the binding between mACE2 and Omicron Spike proteins with measured affinities by BLI up to 14-20-fold stronger compared to Beta (Fig 1A). Interestingly, the BA.1 Q493R mutation eliminates a favorable interaction with the human ACE2 residue K31 due to charge repulsion with R493 and thus is better adapted for binding to mACE2 than to hACE2 (Figs 4B and S10).

## L452Q/R, F486V and the R493Q mutations of Omicron subvariants do not impact mouse or human ACE2 binding

Both the BA.2.12.1 and BA.4/5 Omicron subvariants carry the immune-evading L452Q/R mutation first seen in the Delta variant (S2 Fig). The site of mutation is not within the RBD-ACE2 binding interface and, as expected, the binding affinities for both hACE2 and mACE2 binding to BA.2.12.1 are unchanged compared to BA.2 (Figs 1A, 1B and S9). In the mACE2/BA.2.12.1 structure, the critical interactions made by Spike residues R498 and Y501 with mACE2 within patch 2 are conserved as in BA.1 (Fig 4B). Surprisingly, compared to BA.1, R493 of BA.2.12.1 forms an alternative hydrogen bonding network with mACE2 residues N31 and E35, instead of Q34, that now forms a new hydrogen bond with Y453 of BA.2.12.1 suggesting a plasticity in its interactions (Fig 4B).

The Omicron subvariants BA.4 and BA.5 that were the principal variants during an infection wave in the summer of 2022 also share the L452Q/R mutation. Additionally, BA.4 and BA.5 has gained an additional immunity evading F486V mutation, and the reversion of R493 back to the wild-type Q493 (S2 Fig). The structure of the mACE2/BA.4/5 complex show crucial changes at patch 1, where the BA.4/5 mutation F486V loses Van der Waals interactions with adjacent mACE2 residues F83 and F28 compared to BA.1 (Fig 4B). Q493 on the Spike RBD also no longer forms interactions with mACE2 due to the loss of the longer favorable arginine side-chain.

It has been predicted by modelling and then demonstrated by deep mutation scans of the RBD that the F486V mutation reduces the affinity of the RBD for hACE2 and before the appearance of the BA.4/5 subvariant, mutation at this site was observed at very low rates. [65,66] The hACE2/BA.4/5 complex confirms this loss of hydrophobic contacts for F486V mutation with hACE2 residues F28 and Y83, as seen with mACE2/BA.4/5, however, Q493 now reestablishes a hydrogen bond interaction with hACE2 K31 residue, an interaction previously first lost with the appearance of the Omicron lineage due to the charge repulsion between R498 and K31 (Figs 4C and S10). The combination of these two mutations allows for the BA.4/5 Spike to maintain high affinity interactions with both hACE2 and mACE2. This is an example of the interplay between balancing immune evasion, while maintaining high-affinity receptor binding. The other principal interactions between mACE2 and Spike's BA.4/5 in patch 2 are comparable to those observed between mACE2 and Omicron subvariants BA.1 or BA2.12.1.

## Discussion

New variants of the SARS-CoV-2 virus have emerged posing a challenge to vaccine/antibody efficacy and global pandemic management. We report here that since the appearance of the first variant of concern Alpha, the SARS-CoV-2 Spike glycoprotein has serendipitously gained the ability to also bind the mouse ACE2 receptor potentially broadening its host range.[45–48] Here we demonstrate by a structural approach that this ability to bind the mouse ACE2

receptor has further been maintained even with immune evading mutations in the Omicron lineage.

Our BLI assays with early variants of concern and mutants agree with earlier findings that N501Y and E484K are critical residues that allow for the Spike protein to bind to other mammalian ACE2 homologs. To our knowledge, there has been no reported structure of the Beta Spike bound to a non-human ACE2. Our cryo-EM structure illuminates how a single mutation, N501Y, allows for new interactions especially with the mACE2-unique residue H353 via cation-π interactions, analogous to K353 as observed with Alpha, Beta and Gamma Spike bound hACE2.[67].

With the appearance of the Omicron variant and its subvariants, many epitopes on the surface of the Spike glycoprotein have mutated leading to reduced or abolished antibody binding. The BA.1 variant was first reported with increased resistance to antibodies and significantly stronger binding to hACE2.[60,68] We also observe that mACE2 also shows drastically increased binding (almost 10-fold) to the Omicron BA.1 Spike compared to all other variants that preceded it. The cryo-EM structure of mACE2/BA.1 reveals residues on the Spike protein that increase the number of interactions with mACE2: N501Y, Q493R, G496S and Q498R. This structural observation is consistent with literature reports that reported Spike mutations N501Y, Q493H and K417N increases binding to ACE2 from mice and other species.[34,35,69] We further observe that Spike proteins from Omicron subvariants with significant antibody evasion abilities such as BA.2.12.1 and BA.4/5 still bind to mACE2 demonstrating that the SARS-CoV-2 virus can balance both evasion of the human humoral immune response and its ability to bind to different host species. Notably, this is highlighted at position 493, where in the BA.1 Spike, the glutamine (Q) to arginine (R) mutation allows for new hydrogen bonds with mACE2 residues. However, in humans, it has been shown that this position favored antibody evasion at the cost of an interaction with hACE2 residue K31. Further Omicron subvariants such as BA.4/5 have reverted the arginine back to glutamine while picking up the F486V mutation that reduces van der Waals at the ACE2 binding interface. This compensatory mechanism allowed for further immune evasion while maintaining adequate binding affinity for the hACE2 Spike. Importantly, binding to mACE2 was unaffected even with the loss of two interactions.

Our structural study of human and mouse ACE2 bound to variant Spikes is not without limitations. The ability to bind mACE2 does not necessarily imply that it is able to infect and replicate in mouse.[70] However, numerous *in vivo* mouse studies with early-pandemic, Omicron and mouse-adapted variants demonstrate that these variants are able to infect and cause disease [32,45,46,58,71] In particular, our cryo-EM models shed light on the structural basis of mACE2 binding to these variants. Importantly, our data complements previous studies that also identified mutations that enable mACE2 binding, such as N501Y.

In summary, our structural data and binding analysis of mACE2 to SARS-CoV-2 variants of concern highlight how mutations acquired to evade the immune response in humans may have an impact on the binding to ACE2 receptors of other species thus possibly increasing host susceptibility. Our binding assays and structural analysis identified 4 of the mutations found in SARS-CoV-2 Spikes and known to be associated with immune escape, N501Y, E484A, Q493R and Q498R, as critical mutations involved in high-affinity binding of VOC Spikes to the mouse ACE2 receptor, and as such potentially allowing for expansion of SARS-CoV-2 host range.

## Methods and materials

### Protein production and purification

The Spike trimer was designed to mimic the native trimeric conformation of the protein in vivo and the expression vector was kindly provided by Prof. Jason McLellan, University of

Texas, Austin (TX), USA. It encoded the prefusion ectodomain of the original 2019-CoV Spike, containing a native signal peptide, residues 986 and 987 mutated to proline (2P), a mutated putative furin cleavage site (residues 682–685 mutated to GSAS), a C-terminal T4 foldon fusion domain to stabilize the trimer complex, followed by C-terminal 8x His and 2x Strep tags for affinity purification.[15] The trimeric Spike protein was expressed as previously reported.[72–74] Transiently expressed in suspension-adapted ExpiCHO cells (Thermo Fisher) in ProCHO5 medium (Lonza) at $5 \times 10^6$ cells/mL using PEI MAX (Polysciences) for DNA delivery. At 1 h post-transfection, dimethyl sulfoxide (DMSO; AppliChem) was added to 2% (v/v). Following a 7-day incubation with agitation at 31˚C and 4.5% $CO_2$, the cell culture medium was harvested and clarified using a 0.22 μm filter. The conditioned medium was loaded onto Streptactin XT columns (IBA) washed with PBS and eluted with 50 mM biotin in 150 mM NaCl, 100 mM HEPES 7.5. Eluted protein was then dialyzed overnight into PBS. The purity of Spike trimers was determined to be >99% pure by SDS-PAGE analysis. Point mutations were generated by InFusion-mediated site directed mutagenesis. Variant clones were generated by gene synthesis (Twist Biosciences, Genscript and IDT) on the 2019-CoV Spike background as above. All mutants were produced and purified in an identical manner to the original 2019-Cov S protein.

Human (residues 19–615), mouse (19–615), dog (19–614), cat (19–615) and mink (*neovision vison*) (19–615) ACE2 genes were synthesized by Twist Biosciences and cloned into the pTWIST CMV BetaGlobin WPRE vector. The gene was placed after the human pregnancy specific glycoprotein 1 signal peptide and is followed by a 3C protease cleavage site, a mouse IgG2a Fc fragment and a 10x His tag. Protein production was produced exactly as for the the Spike protein. The filtered conditioned media was then subjected to standard protein A purification. The eluted protein was dialyzed into PBS.

## Cryo-electron microscopy

**mACE2/Beta complex.**   Cryo-EM grids were prepared with a Leica EM GP2 (Leica) plunge-freezing device, using Quantifoil R2/1 copper 400 grids. 3.0 μL of a sample containing 0.4 μM Beta Spike and 0.7 μM mACE2-Fc was applied to the glow-discharged grids, and back-blotted for 2 s with a 10 s wait time, 80% humidity and 10˚C in the sample chamber, and the blotted grids were plunge-frozen in liquid nitrogen-cooled liquid ethane.

Grids were screened for particle presence and ice quality on a TFS Talos Arctica transmission electron microscope (TEM) operated at 200kV. Cryo-EM data was collected using the same microscope, equipped with a TFS Falcon 3 camera. Movies were recorded at a nominal magnification of 150kx, corresponding to a 0.9759Å pixel, with defocus values ranging from -0.8 to -2.5 μm. Exposures were adjusted automatically to 40 e-/Å2 total dose with automatic collection using EPU.

**mACE2/Omicron BA.1 complex.**   Cryo-EM grids were prepared with a Vitrobot Mark IV (Thermofisher Scientific (TFS)). Quantifoil R1.2/1.3 Au 400 holey carbon grids were glow-discharged for 120 s at 15mA using a PELCO easiGlow device (Ted Pella, Inc.). 3.0 μL of a sample containing 9 μM Omicron BA.1 and 16 μM mACE2-Fc was applied to the glow-discharged grids, and blotted for 6 s under blot force 10 at 100% humidity and 4˚C in the sample chamber, and the blotted grids were plunge-frozen in liquid nitrogen-cooled liquid ethane.

Grids were screened for particle presence and ice quality on a TFS Glacios TEM (200kV), and the best grids were transferred to TFS Titan Krios G4 TEM. Cryo-EM data was collected using TFS Titan Krios G4, equipped with a Cold-FEG and Selectris X energy filter, on a Falcon IV detector in electron counting mode. Falcon IV gain references were collected just before data collection. Data was collected using TFS EPU v2.12.1 using aberration-free image shift

protocol (AFIS). Movies were recorded at the nominal magnification of 165kx, corresponding to the 0.726Å pixel size at the specimen level, with defocus values ranging from -0.7 to -2.0 μm. Exposures were adjusted automatically to 60 e-/Å$^2$ total dose.

**hACE2/Omicron BA.4/5, mACE2/Omicron BA.4/5, mACE2/Omicron BA.2.12.1.**
Cryo-EM grids were prepared with a Vitrobot Mark IV (Thermofisher Scientific (TFS)). Quantifoil R1.2/1.3 Au 400 holey carbon grids were glow-discharged for 120 s at 15mA using a PELCO easiGlow device (Ted Pella, Inc.). 3.0 μL of a sample containing 14 μM of the corresponding Spike and 25 μM ACE2-Fc was applied to the glow-discharged grids, and blotted for 6 s under blot force 10 at 100% humidity and 4˚C in the sample chamber, and the blotted grid was plunge-frozen in liquid nitrogen-cooled liquid ethane.

Grids were screened for particle presence and ice quality on a TFS Glacios TEM (200kV), and the best grids were transferred to TFS Titan Krios G4 TEM. Cryo-EM data was collected using the TFS Titan Krios G4 TEM, equipped with a Cold-FEG, on a Falcon IV detector in electron counting mode. Falcon IV gain references were collected just before data collection. Data was collected using TFS EPU v2.12.1 using aberration-free image shift protocol (AFIS). Movies were recorded at nominal magnification of 96kx, corresponding to the 0.83Å pixel size at the specimen level, with defocus values ranging from -0.7 to -2.5 μm. Exposures were adjusted automatically to 60 e-/Å2 total dose.

## Cryo-EM image processing

On-the-fly processing was first performed during data acquisition for evaluating the data quality during screening by using cryoSPARC live v3.3.1.4.[75] The obtained ab-initio structures were used as templates for better particle picking. Motion correction was performed on raw stacks without binning, using the cryoSPARC implementation of motion correction. The full data processing workflow is shown in Supplementary information, S3–S8 Figs and processing statistics on S1 Table.

## Cryo-electron microscopy model building

The models of a SARS-CoV2 Spike (PDB ID 7QO7), mouse ACE2 (PDB ID 7FDK) and human ACE2 (PDB ID 7FDG) were re-fit into the cryo-EM maps with UCSF Chimera.[76] These docked models were extended and rebuilt manually with refinement, using Coot and Phenix. Statistics reported on S1 Table are from MolProbity as implemented in Phenix real-space refine [77,78] Figures were prepared in UCSF ChimeraX.[79]

## Biolayer Interferometry (BLI)

All experiments were performed on a Gator BLI system. Running buffer was 150 mM NaCl, 10 mM HEPES 7.5. For binding assays, dimeric ACE2-Fc was diluted to 10 μg/mL and captured with MFc tips (GatorBio). Loaded tips were dipped into a 3 or 2-fold serial dilution series (300 nM, 100 nM, 33.3 nM, or 100 nM, 50 nM, 25 nM) of target analyte Spike protein. Screening of independent point mutations was done as above at a single concentration of 75 nM. Curves were processed using the Gator software with a 1:1 fit after background subtraction. Plots were generated in Prism 9. Original data is provided in an excel spreadsheet as S1 Dataset.

## Supporting information

**S1 Fig. BLI assays of animal ACE2 against various variants of concern or mutants.**
(TIF)

**S2 Fig. Overview of the domain architecture of selected variants of concern.** Mutations are shown as red lines and labelled for each variant. Specific domains are highlighted: signal peptide (SP), N- terminal domain (NTD), receptor binding domain (RBD), S1 and S2 domain.
(TIF)

**S3 Fig. Cryo-EM processing of the mACE2/Beta complex.**
(TIF)

**S4 Fig. Cryo-EM processing of the mACE2/BA.1 complex.**
(TIF)

**S5 Fig. Cryo-EM processing of the mACE2/BA.2.12.1 complex.**
(TIF)

**S6 Fig. Cryo-EM processing of the mACE2/BA.4/5 complex.**
(TIF)

**S7 Fig. Cryo-EM processing of the hACE2/BA.4/5 complex.**
(TIF)

**S8 Fig. FSC curves indicating resolutions at (FSC 0.143) and final focused refined maps colored by local resolution.**
(TIF)

**S9 Fig. View of the 452 residue in various variants of concern with cryo-EM density.**
(TIF)

**S10 Fig. The BA.1 mutation Q493R abolishes interactions with the hACE2 residue K31 compared to wildtype.**
(TIF)

**S1 Table. Cryo-EM data collection and refinement statistics.**
(DOCX)

**S1 Dataset. BLI Raw data for Figs 1 and S1.**
(XLSX)

## Acknowledgments

Cryo-EM data collection and initial image processing was performed at the Dubochet Center for Imaging, a common initiative of the EPFL, UNIL and UNIGE, with data collection performed in both Geneva and Lausanne. We thank Yashar Sadian for data collection on the Talos Arctica. We thank Rosa Schier, Laurence Durrer, Soraya Quinche and Michael François of the Protein Production and Structure Core Facility of EPFL for helping in the production of Spike Omicron and ACE2 and Charlene Raclot for cloning of the Spike variants.

## Author Contributions

**Conceptualization:** Dongchun Ni, Priscilla Turelli, Florence Pojer, Didier Trono, Kelvin Lau.

**Data curation:** Dongchun Ni, Kelvin Lau.

**Formal analysis:** Dongchun Ni, Priscilla Turelli, Kelvin Lau.

**Funding acquisition:** Henning Stahlberg.

**Investigation:** Dongchun Ni, Bertrand Beckert, Sergey Nazarov, Emiko Uchikawa, Alexander Myasnikov, Henning Stahlberg, Kelvin Lau.

**Methodology:** Dongchun Ni, Bertrand Beckert, Sergey Nazarov, Emiko Uchikawa, Alexander Myasnikov, Kelvin Lau.

**Project administration:** Kelvin Lau.

**Resources:** Priscilla Turelli, Didier Trono, Henning Stahlberg.

**Supervision:** Priscilla Turelli, Florence Pojer, Didier Trono, Henning Stahlberg, Kelvin Lau.

**Validation:** Kelvin Lau.

**Visualization:** Dongchun Ni, Kelvin Lau.

**Writing – original draft:** Dongchun Ni, Kelvin Lau.

**Writing – review & editing:** Dongchun Ni, Priscilla Turelli, Florence Pojer, Henning Stahlberg, Kelvin Lau.

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
