## [Decision Letter · Decision Letter 0]

23 Dec 2022

Dear Dr. Lau,

Thank you very much for submitting your manuscript "Cryo-EM structures and binding of mouse and human ACE2 to SARS-CoV-2 variants of concern indicate that immune escape mutations could expand host range" for consideration at PLOS Pathogens. As with all papers reviewed by the journal, your manuscript was reviewed by members of the editorial board and by several independent reviewers. The reviewers appreciated the attention to an important topic. Based on the reviews, we are likely to accept this manuscript for publication, providing that you modify the manuscript according to the review recommendations.

This is an important structural study on SARS-CoV-2 spike mutations, species tropism and immune evasion. In general, all three reviewers were positive about the manuscript and did not find any major faults with the experiments or data provided. While there are several points to be addressed in the writing, reviewer 1 was primarily concerned with some of the discussion on species tropism and evolution and reviewer 2 had several structural questions regarding trimeric spike conformation. Reviewer 3 noted that this study does not validate the results with functional assays in the form of cell culture or mouse model experiments. The authors should adjust the manuscript with additional context and detail as necessary, as well as provide a thorough point-by-point rebuttal.

Sincerely,

Michael Letko, PhD

Guest Editor

PLOS Pathogens

Sonja Best

Section Editor

PLOS Pathogens

Kasturi Haldar

Editor-in-Chief

PLOS Pathogens

orcid.org/0000-0001-5065-158X

Michael Malim

Editor-in-Chief

PLOS Pathogens

orcid.org/0000-0002-7699-2064

This is an important structural study on SARS-CoV-2 spike mutations, species tropism and immune evasion. In general, all three reviewers were positive about the manuscript and did not find any major faults with the experiments or data provided. While there are several points to be addressed in the writing, reviewer 1 was primarily concerned with some of the discussion on species tropism and evolution and reviewer 2 had several structural questions regarding trimeric spike conformation. Reviewer 3 noted that this study does not validate the results with functional assays in the form of cell culture or mouse model experiments. The authors should adjust the manuscript with additional context and detail as necessary, as well as provide a thorough point-by-point rebuttal.

Reviewer Comments (if any, and for reference):

Reviewer's Responses to Questions

**Part I - Summary**

Reviewer #1: Lau and colleagues present datasets across a panel of relevant SARS-CoV-2 variants for binding affinity to mouse ACE2 and atomic structures of the interactions. As viral surveillance efforts continue to probe for other animals that contract (or potentially propagate) SARS-CoV-2, it is important to have these complementary biochemical studies to understand the future potential of other spillback pathways. Overall, the datasets are useful. I’m not ordinarily so nitpicky, but I found myself noting a lot of small aspects of writing and scholarship that could be improved, but these are minor comments and I do not have any major critiques on the study details or the datasets that are presented in the manuscript.

Reviewer #2: Ni et al. analyzed the structures of SARS-CoV-2 VOC mutants bound to mouse or human ACE2, and measured the corresponding dissociation constants. The authors claimed that mutations promoting immune escape could also help SARS-CoV-2 achieve cross-species transmission. Note that previous studies have proposed the possibility of a mouse origin of Omicron based on computational analyses; this paper provided critical experimental data to improve our understanding of the evolution of Omicron and therefore would attract a broad group of readers interested in the evolution of SARS-CoV-2, especially the Omicron variant. I found the paper generally easy to follow and would be happy to recommend its publication in PLoS Pathogens after addressing the following questions.

Reviewer #3: This manuscript address the potential concerns of the spillover of SARS-CoV-2 to mouse to establish a new reservoir using the structural investigation. The results give comprehensive and detailed mechanisms for the enhanced binding capacity of the VOCs to mouse ACE2. The main weakness is the lack of confirmation using the functional assays such as cell infectivity of mouse model infections.

**Part II – Major Issues: Key Experiments Required for Acceptance**

Reviewer #1: none

Reviewer #2: (No Response)

Reviewer #3: The cell infectivity or mouse model assays could be used to verify the conclusions.

**Part III – Minor Issues: Editorial and Data Presentation Modifications**

Reviewer #1: Line 21-22 – I don’t find the argument that mice may act as reservoirs for SARS-CoV-2 compelling. SARS-CoV-2 doesn’t need a reservoir, it’s an endemic human virus.

Lines 39-41 describe some uncertainty about ultimate reservoir host of these viruses, but I don’t think there’s any doubt that the natural reservoir is Rhinolophus bats (as opposed to “some evidence” suggested in line 41).

Citation 16 is likely meant to be a different Walls et al. paper (Cell 2020)?

One potentially helpful part of context outlined in the introduction (e.g. around line 52), is that many of the spike residues that are dominant targets of neutralizing immunity are also those that influence ACE2 binding specificity and therefore potentially host tropism. Therefore, turnover of amino acids at these positions due to antigenic selection in humans could have the secondary effect of influencing potential host range. This logic is implicit in what I believe the authors are saying, but it could be laid out more specifically in the introduction.

Nomenclature and being consistent – e.g. just between lines 66 and 68, BA4/5 and BA 4/5 (with and without space) is used. I believe the correct nomenclature for the Omicron variants has a period in it (e.g., BA.4/5), and I usually see the Greek letters (e.g. Beta) capitalized

Line 118: “required” implies that no other route toward mACE2 binding is possible in the absence of N501Y, which disagrees with other publications that show other ways of acquiring mACE2 binding independent of N501Y (e.g., PMID 32854108, 35114688). Just because naturally evolved SARS-CoV-2 variants to date have acquired mACE2 binding via N501Y doesn’t mean that mutation is “required” in a global sense.

Line 134: in e.g. BA.1, 15 of 34 mutations being in the RBD is precisely not a “majority”

Line 172: “Y453 of mACE2” – Y453 is a spike residue, not a mACE2 residue

Line 183: Why point to molecular modeling for the claim that F486V is affinity-decreasing when it has been shown experimentally? (e.g., https://doi.org/10.1371/journal.ppat.1010951).

Line 220: “… N501Y … also increases binding to mouse-adapted strains of SARS-CoV-2” – some sort of grammatical/logical error here (I think it should be, “also increases binding to mouse ACE2 (and other potential host species) in mouse-adapted strains of SARS-CoV-2” or something to that effect

Throughout, over-reliance on ACE2 binding = potential reservoir. Being a reservoir is about more than receptor binding, it depends on ecology and animal population dynamics and viral restriction versus permissivity and transmission and so many other factors. Language therefore needs to be tightened up in this regard, e.g. lines 21, 200-201, 225. In contrast, an appropriately agnostic statement as a good example is line 237-238.

Reviewer #2: 1. The title includes the phrase “expand host range”, which is also present in Discussion (Lines 241-242). However, the only host other than humans tested in the study is mice. It is probably better to specify the alternative host in this case, unless the authors would like to provide additional data about the ACE2-spike interactions in other host species.

2. I have doubts if Fig. 1c was correctly cited in the main text (lines 164-165). The difference in amino acid sequences is provided in Fig. 1c but is not explicitly discussed in the main text.

3. The authors should explain in more detail about the 2-up and 3-up conformation to the broad audience.

4. The authors compared their structure to PDB: 6M0J, which if I understood correctly represents interactions of monomers between spike RBD and ACE2. I wonder how the trimeric spike and dimeric ACE2 investigated in this study should result in identical structures as those solved from monomeric protein domains. Is this topic worth some elaborated discussion/explanation during revision?

5. I would suggest the authors introduce more information about the 2-up and 3-up conformations of ACE2. They appeared in lines 101-102 and supplementary figures without sufficient context. And why did only 2-up show up in the mACE2+Beta structure (Fig. S2)?

6. Figure 3c in the legend should be Figure 3b.

7. It is not clear how structures in Figure 2c and Figure 3b were aligned. In my view the authors should align mACE2 (or hACE2) for all structures, so that we can see how the Omicron variants structurally varied.

8. Back to the title, the authors did not really show that all these mutations of Omicron variants were associated with immune escape in this study. Probably some indeed contributed to immune escape while others might not. Therefore, I am not sure if it is appropriate to have them in the title.

Reviewer #3: The Figure 2 and Figure S9 could not totally align. In figure 2, the binding capacity to mACE2 for the VOC is wildtype<beta<alpha<gamma<omicron, figure="" in="" s9="" while=""></beta<alpha<gamma<omicron,>

PLOS authors have the option to publish the peer review history of their article (what does this mean?). If published, this will include your full peer review and any attached files.

Reviewer #1: No

Reviewer #2: No

Reviewer #3: No

Figure Files:

Data Requirements:

Reproducibility:

References:

---

## [Editor Report · Decision Letter 1]

13 Feb 2023

Dear Dr. Lau,

We are pleased to inform you that your manuscript 'Cryo-EM structures and binding of mouse and human ACE2 to SARS-CoV-2 variants of concern indicate that mutations enabling immune escape could expand host range' has been provisionally accepted for publication in PLOS Pathogens.

Best regards,

Michael Letko, PhD

Guest Editor

PLOS Pathogens

Sonja Best

Section Editor

PLOS Pathogens

Kasturi Haldar

Editor-in-Chief

PLOS Pathogens

orcid.org/0000-0001-5065-158X

Michael Malim

Editor-in-Chief

PLOS Pathogens

orcid.org/0000-0002-7699-2064

Thank you for thoroughly addressing each of the reviewers' comments and revising your manuscript. The text now carefully navigates the specific language in the field while remaining accessible to a broader audience. As the requested comments were minor and no additional data was included in the revisions, we have not sought additional review and now find the manuscript suitable for publication in PLoS Pathogens.
---

## [Editor Report · Acceptance letter]

30 Mar 2023

Dear Dr. Lau,

We are delighted to inform you that your manuscript, "Cryo-EM structures and binding of mouse and human ACE2 to SARS-CoV-2 variants of concern indicate that mutations enabling immune escape could expand host range," has been formally accepted for publication in PLOS Pathogens.

Best regards,

Kasturi Haldar

Editor-in-Chief

PLOS Pathogens

orcid.org/0000-0001-5065-158X

Michael Malim

Editor-in-Chief

PLOS Pathogens

orcid.org/0000-0002-7699-2064